# Withdrawing Antipsychotics for Challenging Behaviours in Adults with Intellectual Disabilities: Experiences and Views of Prescribers

**DOI:** 10.3390/ijerph192417095

**Published:** 2022-12-19

**Authors:** Gerda de Kuijper, Joke de Haan, Shoumitro Deb, Rohit Shankar

**Affiliations:** 1GGZ-Drenthe/Centre for ID and Mental Health, Middenweg 19, 9404 LL Assen, The Netherlands; 2Academic Collaboration ID and Mental Health, Department Psychiatry and Department Family Practice, University Medical Centre Groningen, 9713 GZ Hanzeplein, The Netherlands; 3Department of Brain Sciences, Faculty of Medicine, Imperial College London, Du Cane Road, London W12 0NN, UK; 4Peninsula School of Medicine, University of Plymouth and Cornwall Partnership NHS Foundation Trust, Plymouth PL4 8AA, UK

**Keywords:** intellectual disabilities, antipsychotics, challenging behaviour, discontinuation, prescribers’ policies, survey

## Abstract

International current best practice recommends the discontinuation of antipsychotics for challenging behaviours in people with intellectual disabilities (ID), due to lack of evidence of efficacy and risks of harmful side-effects. In clinical practice, discontinuation may be difficult. The aim of this study was to gain insight into prescribers’ practice by investigating their experiences with the discontinuation of long-term antipsychotics for challenging behaviour. From professionals’ associations thirty-four registered ID physicians, psychiatrists and specialist mental healthcare nurses were recruited who completed an online questionnaire in this survey–study. Almost all participants had attempted to deprescribe antipsychotics for their patients with ID. Sixty-five percent of participants achieved complete discontinuation in 0–25% of their patients, but none in over 50%. Barriers were a lack of non-pharmaceutical treatments for challenging behaviours and caregivers’ and/or family concern. Seventy percent of participants indicated that their institutions had encouraged implementing their discontinuation policies in line with the new Dutch Act on Involuntary care and a new Dutch multidisciplinary guideline on problem behaviour in adults with ID. Support and facilitation of clinicians from institutions’ managers and political and professional bodies may be helpful in further implementation of best practice in the treatment of challenging behaviour in people with ID.

## 1. Introduction

The rates of challenging behaviours displayed by people with intellectual disabilities (ID) vary from 18% in community populations to 85% in people with profound ID who live in institutions [1,2,3]. The onset and maintenance of challenging behaviours in individuals with ID are associated with biological, psychological and environmental factors [1,2,4,5,6]. Behaviours include outwardly directed behaviours like aggressive-destructive behaviour, inappropriate (sexual) behaviour and disruptive behaviours. It also includes hyperactivity, irritability, lethargy and self-injurious, stereotypic and withdrawn behaviours [2,7,8]. Challenging behaviours may vary in intensity and be persistent [9,10,11,12]. When individuals present with challenging behaviour, this may negatively influence their quality of life [13,14] and social participation [15]. Diagnosis and management require a multidisciplinary and integrative approach as the causes of these behaviours are multi-factorial (physical, genetic, psychiatric, psychological, social, environmental, etc.) [3,6,16].

Psychotropic drugs, especially antipsychotics, are often prescribed for the management of challenging behaviours [17,18,19,20,21,22]. Significantly higher levels of antipsychotic drug prescribing occur to manage challenging behaviour in people with ID and/or autism [23]. However, the effectiveness of psychotropic drugs for the treatment of challenging behaviours in people with ID has not been proven [24,25,26,27]. Moreover, in this population, the risk of side-effects like diabetes, cardio-vascular disorders, sedation and movement disorders is considerable, especially in the case of long-term use of antipsychotics [28,29,30,31]. It is recommended by all good practice guidelines that the treatment of challenging behaviours should be, in the first place, non-pharmacological, i.e., psychosocial and behavioural. Practitioners should only prescribe psychotropic drugs after careful weighing of the potential positive and negative effects of this medication and preferably just for the short-term (max. 6–12 weeks). For example, prescription may be beneficial to overcome crisis situations when the person or others are at risk of serious harm, or in conjunction with non-pharmacological interventions [16,32,33].

Although it is recommended that long-term off-label psychotropic drug prescription should be discontinued especially antipsychotics, they are still frequently prescribed for the treatment of challenging behaviours [34,35,36]. This may, at least partially be, because of failures of successful discontinuation of antipsychotics due to worsening of behaviour upon drug withdrawal [37]. Besides user-related factors, like the occurrence of withdrawal symptoms, symptoms of previously not-recognized mental or physical disorders or other health problems, and staff-related factors, are associated with unsuccessful withdrawal of antipsychotics [38]. For example, staff may experience difficulties in the management of changes in their clients’ behaviour, may have negative feelings towards clients’ behaviour or lack knowledge or have unrealistic expectations about the effects of psychotropic drugs [39,40]. The involvement of all stakeholders, including the patients and their relatives, seems a key-element in the achievement of successful changes in prescription policies [41].

In some countries, initiatives have been taken to facilitate the deprescribing of antipsychotics for challenging behaviours. For example, in the UK, the “stopping over medication of people with a learning disability, autism or both with psychotropic medicines” (STOMP) is now incorporated within the NHS England long-term plan [42]. The development of a structured pathway with the involvement of all stakeholders led to successful discontinuation in 46% of cases and dose reductions of over 50% in another 11% [43]. A survey among UK psychiatrists about changes in prescription policies as a result of the STOMP initiative showed that the prescriptions practice has changed. The UK psychiatrists are less likely to initiate new prescriptions of antipsychotics. However, complete antipsychotic discontinuation in over 50% of their patients was achieved by only 4.5% of responding psychiatrists [44].

In The Netherlands, Vilans, a knowledge centre on long-term care has conducted a project (2015–2019) to reduce inappropriate psychotropic prescribing in the elderly with dementia and in people with ID and high care needs. In this project, an e-learning on psychotropic drug use in people with ID for support professionals has been developed [45]. Moreover, in The Netherlands, in December 2019, a new multidisciplinary guideline on challenging behaviour [46] and, in January 2020, a new Dutch Act on Involuntary care was published. In both, the prescription of psychotropic drugs for challenging behaviours outside guideline recommendations is regarded as restricted practice. A study among Dutch ID physicians on their experiences with the guideline and new Act showed that they regarded the Act as supportive of reducing the off-label prescriptions of psychotropics. However, because of organisational difficulties, the feasibility of implementing the guideline and new Act is not always possible in practice [47].

In The Netherlands, there are 250 registered ID physicians. Most of them take care of people with ID and complex needs, including people with multiple disabilities and/or behavioural problems.

There are approximately 15–20 ID specialized mental healthcare outpatient clinics spread over The Netherlands. Psychiatrists in mainstream mental healthcare organizations or who have their own private practice may also be consultants at ID institutions. In Dutch mental healthcare, specialist nurses monitor the use of the psychotropic drugs and are authorized to stop or change a prescription in consultation with the psychiatrist who initiated the prescription. In community care, GPs may provide repeat prescriptions and/or the monitoring of psychotropic drug use; however, mostly, they refer to ID physicians or specialized ID mental healthcare when discontinuation or change in medication is necessary. The total number of mental healthcare professionals affiliated with the Dutch ID mental healthcare outpatient clinics, authorized to prescribe psychotropic medication, is unknown. We estimate 100–150 such prescribers based on the number of ID mental healthcare clinics and consultant -psychiatrists specializing in people with ID in The Netherlands.

The present study aimed to gather information about facilitators and barriers to the successful discontinuation of antipsychotic drugs when used for challenging behaviours in people with ID. We conducted a survey, similar to the UK survey [44], about experiences with antipsychotic drug discontinuation among ID physicians, psychiatrists and specialist nurses in The Netherlands who were responsible for the psychotropic drug prescriptions for their patients with ID. We also asked whether their antipsychotic prescribing practice had changed because of the new guideline and new Act.

## 2. Materials and Methods

### 2.1. Design

This was a cross-sectional study including a questionnaire survey among Dutch prescribers of psychotropic medications for people with ID.

### 2.2. Participants and Settings

Participants were registered ID physicians affiliated with ID institutions and institutions’ outpatient clinics or having their own practice, and psychiatrists or specialist nurses affiliated with ID-specialized mental healthcare outpatient clinics or having their own practice.

### 2.3. Materials

The questionnaire items were adapted for the Dutch ID healthcare system from a survey which was developed in a recent study of UK psychiatrists’ views on the rationalization of psychotropic use among people with ID [44]. The survey answer options of items were multiple choice with an open field for comments. Appendix A shows the items of the questionnaire and corresponding answer options. The forty-two questionnaire items were grouped under five themes similar to those in the UK survey. These themes included:(1)Data on working experience and antipsychotic prescription practice in patients with ID;(2)Experience in deprescribing antipsychotics;(3)How their antipsychotic prescribing practice including deprescribing, changed over the past five years;(4)How in the past five years new Dutch Acts and guidelines influenced antipsychotic deprescribing;(5)Potential facilitators and barriers in achieving the goal of deprescribing.

### 2.4. Procedures

The participants were recruited from the ID physicians affiliated with the Dutch Intellectual Disabilities physicians’ association (NVAVG), psychiatrists affiliated with the Dutch Psychiatrists Association (NVvP) and nurses affiliated with the Dutch specialist nurse care association (VenVNVS). We had to approach all the Dutch psychiatrists and specialist nurses as there is no Register of those who specifically specialize in ID mental healthcare.

The survey was open between 19 February and 1 May 2021. A reminder was sent three weeks before the closing date. A news item with the link to the questionnaire was also placed on the websites of the professionals’ associations. Members of the professional organisations/potential responders received an explanatory letter with a link to the program Qualtrics where they could fill in the questionnaire. On average, it took participants 30–45 min to complete the questionnaire.

### 2.5. Ethics

All responders in the survey participated voluntarily. All data were collected, stored, and safeguarded anonymously, according to the European Act on Protection of Personal Information, in The Netherlands, which was ratified in 2018.

### 2.6. Analyses

SPSS 26 was used to analyse the survey data. Descriptive statistics were used to calculate frequencies of respondents’ characteristics and answer categories as per the multiple-choice questions. Pearson Chi-Square was used to compare characteristics and answer categories between a group consisting of ID physicians on the one hand and psychiatrists and ID specialist mental healthcare nurses on the other.

## 3. Results

We present the results by the forming of two clusters encompassing the five themes of the questionnaire as described under Methods/Materials (themes 1 and 2, and themes 3, 4 and 5 respectively). The data and analyses of all the 42 questions and corresponding answers (see Appendix A) are available on request.

### 3.1. Participants and Prescription Patterns (Themes 1 and 2 of the Questionnaire)

Twenty-one of the 250 ID physicians registered in The Netherlands and thirteen of the estimated 100–150 mental healthcare professionals prescribing psychotropics to patients with ID (i.e., psychiatrist or specialist mental healthcare nurse) completed the questionnaire (response rate of 8.5% and estimated response rate 8.6–13%, respectively). The exact response rate of the ID mental healthcare professionals could not be calculated because there are no data on the total number of those in ID mental healthcare in The Netherlands.

Twelve (35%) of the total 34 participants reported that their working experience in the field was for less than 10 years, 10 (30%) had worked in the field for 10–19 years, 11 (32%) 20–29 years, and 1 (3%) for more than 30 years. Five (15%) of the participants stated that 0–25% of their patients, 17 (50%) that 25–50%, 5 (14%) that 50–75% and 4 (12%) that 75–100% of their patients with ID were prescribed antipsychotics. Three participants (9%) could not provide the percentage.

### 3.2. Participants’ Policies and Experiences (Themes 3, 4 and 5 of the Questionnaire)

Table 1 shows the results of participants’ practice and experience regarding the deprescribing of antipsychotics in their patients with ID. Participants attempted to deprescribe antipsychotics in a variable part of their patients. In Table 1, the number and percentage of participants is shown who attempted to deprescribe antipsychotics in a certain proportion of their patients.

A considerable number of participants did not attempt to withdraw antipsychotics in their patients with ID and challenging behaviour when there was a clear diagnosis of a psychiatric disorder which justified the prescription. This diagnosis was confirmed recently (i.e., in the two preceding years) in more than half of the total number of respondents (55%). A significantly lesser proportion of ID physicians (38%) compared with ID mental healthcare professionals (ID psychiatrists and nurses) (83%) confirmed the psychiatric diagnosis (*p* = 0.01). Only eight (24%) prescribers confirmed psychiatric diagnosis themselves (ID physicians 10%, ID healthcare professionals 50%, the difference not significant). In addition, there were differences between these two groups with regard to the mean duration of the discontinuation trajectory, which was, on average, significantly longer in ID physicians (0–6 months, 9% in ID physicians versus 8% in ID mental healthcare professionals, 6–12 months, 24% versus 77%, 12–18 months, 47% versus 0%, and >18 months in 19% versus 15%, respectively, *p* = 0.01)

Table 2 shows data which complicated the withdrawal process of antipsychotics for challenging behaviour.

Most participants stated that the antipsychotic drug had to be re-instated in some patients. Furthermore, behavioural worsening or other complications like withdrawal dyskinesia or re-emergence of sleep problems or irritability could occur, leading to delay in the discontinuation process. In addition, staff could be unable or need extra time to adapt their guiding style, or to change clients’ environment or daily activities when clients became less sedated and more active because of the antipsychotic’s withdrawal. These environmental factors could also lead to delay or sometimes failure in the discontinuation process. Nevertheless, complete withdrawal was also achieved in some of the patients, even those whose behaviour deteriorated upon dose reduction.

### 3.3. Factors Related to Deprescribing Antipsychotics in the Working Field of Participants

Twenty-three (70%) of the thirty-four participants indicated that the institutions they were affiliated with had encouraged their clinicians to reduce antipsychotic drugs because of the new Dutch Act on Involuntary care, which is aimed to reduce restrictive practice and the new Dutch guideline on problem behaviour in adults with ID. They also indicated that they were less likely to initiate antipsychotics for challenging behaviours.

Participants also mentioned some factors that facilitated successful discontinuation and barriers that hindered this process. For example, it was difficult to achieve complete withdrawal among those who received polypharmacy of antipsychotics. On the contrary, the concurrent use of other psychotropics (polypharmacy of psychotropics rather than antipsychotics) facilitated complete withdrawal. Other facilitators were the presence of side effects, first attempt to withdraw, and a positive attitude and involvement of clients’ families and support staff. Barriers were resistance against discontinuation from support professionals or family, which was mentioned by 26 (76%) and by 25 (74%) participants, respectively, and unavailability of non-pharmaceutical treatments for clients’ challenging behaviours, which was mentioned by 20 (59%) participants. In addition, lack of knowledge about the discontinuation process, e.g., time schedules and steps in dose reductions (20 participants, 59%), and lack of multidisciplinary team involvement (8 participants, 24%) were mentioned.

## 4. Discussion

In this study, we explored the experiences and views of prescribers regarding withdrawal of long-term antipsychotic drug use for challenging behaviours in people with ID in The Netherlands. Although the long-term prescription of antipsychotics for challenging behaviours in the absence of a psychiatric illness is not recommended [16,46], we found that approximately half of the 21 ID physicians and 13 psychiatrists/specialist mental healthcare nurses who completed the survey questionnaire were still prescribing antipsychotics in the absence of a psychiatric diagnosis in 0–25% of their patients. However, all ID physicians and all but one of psychiatrists/mental healthcare nurses had attempted to deprescribe antipsychotics in the preceding five years. This percentage is higher than in the survey of Deb et al. [44], who found that about half of their 88 respondents had attempted to deprescribe antipsychotics for challenging behaviour in their patients with ID. We found a similar percentage to Deb et al. [44] (65% versus 60% of participants) who succeeded in complete withdrawal in 0–25% of their patients. However, none of the participants in our study was successful in completely discontinuing antipsychotic medications in over 50% of their patients within 12 months compared to 4.5% of participants in the study of Deb et al. [44]. Our participants stated that objections and concerns of family and professional caregivers, and unavailability of non-pharmaceutical treatments and multidisciplinary team input, were barriers in starting or continuing a withdrawal process.

Deb et al. [48] recently completed a qualitative analysis of free text data returned by the UK psychiatrists’ during the questionnaire survey [44] and found that the psychiatrists in the UK like the participants in the current Dutch study found that the lack of resources such as multidisciplinary team input and lack of caregiver support for the withdrawal hampered their attempt to deprescribe antipsychotics. As far as we know, there are no studies from other countries reporting prescribers’ experiences with antipsychotics discontinuation in patients with ID.

As described above and known from other studies, involvement of all stakeholders in decisions about the withdrawal of long-term used antipsychotics for challenging behaviours is a key factor for success [43,44]. Besides organizational barriers and time-constraints in prescribers’ attempts to apply the deprescribing policies [44], staff’s attitudes and knowledge around psychotropic drug use are important factors in outcomes of discontinuation trajectories [40]. However, the most important factor may be the involvement of family-caregivers and patients with ID themselves. Studies have shown that family-caregivers, although they have knowledge on the challenging behaviour of their relative and would like to be heard, are often not involved in treatment decisions, e.g., psychotropic drug prescribing [41]. In addition, often individuals with ID themselves lack information and/or their voices are not heard in decisions in prescription of antipsychotics [41,49]. Previous studies of interviews of adults with ID have just touched on the issue of psychotropic medication use [50,51]. To address this knowledge gap, in another study, we also gathered information on the experience of adults with a mild ID who have gone through the experience of psychotropic withdrawal [52]. An important finding was their statement that their own coping style, next to a supportive environment and a good relationship with their doctor, was a key factor for successful discontinuation.

Legal issues may play a role. We found that 70% of the participants in our study indicated that the new Dutch Act on Involuntary care, aimed to reduce restrictive practice, and the new Dutch guideline on problem behaviour in adults with ID had encouraged their service providers to facilitate psychotropic drugs deprescription. However, clinicians are allowed to deviate from this Act and from the Dutch guideline on problem behaviour provided that they can substantiate this decision. Since the present study was the first in The Netherlands on the issue of experiences of prescribers in withdrawing antipsychotics for challenging behaviours of people with ID, it is difficult to know definitely about the possible facilitating effect of these government Act and guidelines, for which future studies are needed.

Other contextual factors also may play a role in our study, since we found differences in antipsychotics discontinuation policies between ID physicians and ID mental healthcare professionals, who have different working settings. ID physicians took significantly more time for the discontinuation process, possibly because their patients’ stakeholders’ networks are more complex, and it will take more time to coordinate all stakeholders’ interests. Another explanation for the longer discontinuation trajectory may be the difference in confirmation of the presence of a psychiatric illness in patients who started the withdrawal process. This was significantly less often confirmed by ID physicians compared with ID mental healthcare professionals. The lack of confirmation will likely lead to uncertainty and fear of family, caregivers and clinicians regarding re-occurrence of maladaptive behaviour or psychiatric symptoms, and delay in the withdrawal process.

Our study has some limitations. First, the response rate of our survey was low. Approximately 10% of ID physicians had completed the survey questionnaire, and we estimated a similar percentage of the ID mental healthcare professionals (psychiatrists and specialist nurses working in ID mental healthcare and/or affiliated with ID institutions). We were not able to calculate the exact response rate of the latter group because there is no data on the number of those professionals. Therefore, the results of our survey may not be representative of all ID physicians, ID psychiatrists and ID specialist nurses in The Netherlands. However, the low response itself raises some interesting observations. A similar survey conducted two years ago in the UK with the same equivalent target audience i.e., psychiatrists who work with people with ID had a response rate of approximately 40%. This raises questions on the differences in attitudes of the prescribers towards this challenging topic. There could be complex confounders, such as time, resource, training, education, knowledge acquisition and cultural attitudes, which might have influenced these response rates. It could be that a different methodology such as face to face interviews might result in better response rates and better understanding of the barriers to the current survey response. However, such projects can be time consuming and resource intensive.

Furthermore, because this study was conducted in Dutch healthcare settings, the results cannot be generalized to other countries. However, it is known that, in many countries, prescribers encounter the same problems in deprescribing antipsychotics in their patients with ID [37]. Therefore, the results of this study may also be applicable to other Western countries.

## 5. Conclusions

In line with the available international guidelines and a new Dutch Act on Involuntary care, most of the thirty-four participants in our survey of ID physicians, psychiatrists and specialist mental healthcare nurses in The Netherlands had attempted to discontinue their long-term prescriptions of antipsychotics for challenging behaviours in patients with ID. However, complete discontinuation could be achieved in only a small proportion of cases. Concerns of relatives and care and support staff, lack of non-pharmaceutical treatments and lack of multidisciplinary teams’ input were perceived as barriers to successful antipsychotic withdrawal attempts.

Besides improvement in the availability of multidisciplinary teams and non-pharmaceutical treatments, education of care-professionals in supporting clients in the management of medication use is recommended.

Facilitation and support from managers at the institutional level, and creating conditions at the political and professional level, seem to be prerequisites in the successful implementation of deprescribing antipsychotic medication for challenging behaviours in patients with ID.

## Figures and Tables

**Table 1 ijerph-19-17095-t001:** Practice and experience of intellectual disability (ID) physicians and ID mental healthcare professionals ^#^ who completed the survey (N = 34 participants) in deprescribing antipsychotics for challenging behaviours.

Policies and Experiences of Participants	Percentage of Those Patients with ID on Antipsychotics	Participants Responding to Survey (Number, %)
Attempts to withdraw antipsychotics	0%	0 (0%)
	1–25%	3 (9%)
	26–50%	9 (26%)
	>50%	18 (53%)
	missing	4 (12%)
Achieved complete withdrawal	0%	3 (9%)
	1–25%	22 (65%)
	26–50%	5 (14%)
	>50%	0 (0%)
	missing	4 (12%)
Achieved dose reduction of > 50%	0%	0 (0%)
	1–25%	18 (53%)
	26–50%	9 (26%)
	>50%	3 (9%)
	missing	4 (12%)
Not attempted to withdraw because of licensed indication/clear psychiatric diagnosis	0%	2 (6%)
	1–25%	13 (38%)
	26–50%	9 (26%)
	>50%	6 (18%)
	unknown	4 (12%)

^#^ Psychiatrists and specialist mental healthcare nurses affiliated with ID mental healthcare organizations.

**Table 2 ijerph-19-17095-t002:** Complications in withdrawing antipsychotics for challenging behaviour in patients with intellectual disabilities as experienced by participants in a survey study (N = 34).

Complications	Proportion of Patients with the Complication	Participants (n, %) Reporting This Complication
Antipsychotic had to be reinstated within - 3 months		14 (41%)
	0%	4
	1–25%	7
	26–50%	1
	>50%	2
Antipsychotic had to be reinstated within - 6 months		17 (50%)
	0%	2
	1–25%	11
	26–50%	4
	>50%	0
Antipsychotic had to be reinstated within - 12 months		25 (74%)
	0%	3
	1–25%	12
	26–50%	6
	>50%	4
	unknown	0
Occurrence of severe behavioural worsening or severe worsening of mental condition	0%	4 (12%)
	1–25%	19 (56%)
	26–50%	6 (18%)
	>50%	1 (3%)
	unknown	4 (12%)
Delay in discontinuation process due to behavioural worsening after complete withdrawal	0%	4 (12%)
	1–25%	13 (38%)
	26–50%	4 (12%)
	>50%	3 (9%)
	unknown	10 (29%)
Delay in discontinuation process due to behavioural worsening after incomplete withdrawal	0%	0 (0%)
	1–25%	8 (24%)
	26–50%	12 (36%)
	>50%	5 (15%)
	unknown	8 (24%)
Complications (multiple answers allowed): Mild behavioural worsening Withdrawal symptoms Re-emergence of psychiatric symptoms Contextual problems		29 (35%)
		19 (23%)
		24 (29%)
		10 (12%)
Delay in discontinuation process due to complications/complete withdrawal	0%	5 (15%)
	1–25%	12 (35%)
	26–50%	8 (24%)
	>50%	1 (3%)
	unknown	8 (24%)
Delay in discontinuation process due to complications/incomplete withdrawal	0%	0 (0%)
	1–25%	11 (32%)
	26–50%	9 (26%)
	>50%	5 (15%)
	unknown	9 (26%)

## Data Availability

The data are available on request and with a substantiated explanation at GGZ Drenthe/department research https://ggzdrenthe.nl/research, (accessed on 14 December 2022).

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
