# Peer review of "Withdrawing Antipsychotics for Challenging Behaviours in Adults with Intellectual Disabilities: Experiences and Views of Prescribers"

_ijerph, 2022, doi:10.3390/ijerph192417095_

Round 1
Reviewer 1 Report
This manuscript reports the results of a small survey study in which the objective was to determine whether healthcare providers responsible for the treatment of intellectually-disabled patients successfully adhered to recommendations that antipsychotic medications should not be relied upon to counter “challenging behaviors.” In general, the findings indicated that this recommendation is rarely followed and in cases where it is (or was), the success rate is dismal. The authors offer several reasons that potentially explain implementation failure and offer recommendations for future success.
I am not recommending publication of this paper largely because the actual/estimated response rate was between 8.5 and 13%, and this yielded a sample size of only 34, a number that almost certainly cannot be representative of the target population. Rather than attempting to publish these results, I feel the authors should have interpreted the low response rate as a rationale for a better sampling strategy such as face-to-face interviews. As a point of reference, although admittedly somewhat conservative, Bartlett, Kotrlik, & Higgins (2001), in "Organizational Research: Determining Appropriate Sample Size in Survey Research;" Information Technology, Learning, and Performance Journal; Vol. 19, No. 1; the authors recommend 123 to 207 respondents for a population size of 300 (this would be equivalent to response rates of 41% to 69%). Draugalis and Plaza (2009) recommend a rate of 80%. In general, response rates below 30% are rarely considered acceptable especially when this would translate to a sample size of fewer than 100 participants.
In addition to the issue noted above, the manuscript suffers from other deficits that also make its publication worthiness questionable.
1. The Introduction section should be streamlined so that the authors express the main points in a more concise manner. For the topic at hand, a single page should be sufficient.
2. The manuscript overall should be edited by a native speaker of the English language to correct numerous awkward or incomplete sentences, repetitive phrases (the terminology “challenging behaviors” appears 37 times in this paper), and incorrect punctuation (i.e., the use of semicolons rather than commas in sentences consisting of an a, b, c series). On a related note, the authors should have enumerated the five (i.e, 1, 2, 3, 4, 5) mentioned in the Methods section rather than alphabetizing them. On another related note, for the sake of consistency, the authors should have provided an acronym for the Dutch Psychiatrists Association since they did so for the Dutch Intellectual Disabilities physicians’ association (NVAVG) and the nurses affiliated to the Dutch specialist nurse care association (VenVNVS).
3. The authors should clearly define the “side-effects” that are causing concern, and they should also point out that a cost-benefit analysis is always required to weigh the negative effects of a medication against its positive effects prior to prescribing. Many medications other than antipsychotics are prescribed off-label due to an acceptable cost/benefit analysis.
4. The entire section entitled “Setting” does not belong in the Methods section. Typically, the components of a methods section are Participants, Materials, and Procedures. The details about the training of ID specialists (presently located in the Methods section) does not seem particularly relevant to the present study, but assuming that it is, it should be addressed either in the Introduction or Discussion.
5. The authors should explain why they consider it “surprising” that patients’ own coping styles, the existence of supportive environments, and a good relationship with their doctor, was a key factor for successful discontinuation. It seems to me that this finding should have been expected.
6. The Discussion section should be significantly abbreviated so that the main points are more concisely discussed. As an example, the detailed description of the work by Deb et al. is unnecessary to make major points about the findings within the present investigation.
In conclusion, I recommend the authors attend to the issues noted above, consider their present effort a pilot study, and use these result as a rationale to run a more comprehensive assessment of their topic prior to seeking publication. Thank you.
Author Response
Dear Editor and Reviewers,
We thank you for your comments and questions which we have tried to address and answer as best as possible. We think that the clearness and readability of the manuscript has been improved by following the reviewers’ suggestions. The quality of the manuscript has improved by the shortening of the introduction and discussion section, a better explanation and lay-out of the tables and removing and adding of some text fragments. We have also paid attention to the writing style, which we hope now comes across as being better.
Please find below our specific replies to the reviewers’ comments.
Reviewer 1
Reviewer 1 does not recommend publication of the manuscript, largely because of the low response rate to our survey. The reviewer suggests that this low response rate should be a reason to use another sampling strategy which could have been more effective, i.e., face-to-face interviews.
Answer:
We accept the reviewer’s criticism of a low turn out to our survey. This is a valuable consideration. We would like to highlight the fact that response rates to medical professional surveys is generally low (20-30%). It is interesting to note that the same survey using similar design and questions carried out in the UK a couple of years ago appears to have had a better response rate approximately 40% compared to an estimated 10% to this survey. We have discussed this briefly in the limitations as follows:
‘However, the low response itself raises some interesting observations. A similar survey done two years ago in the UK with the same equivalent target audience i.e. psychiatrists who work with people with ID had a response rate of approximately 40%. This raises questions on the differences in attitudes of the prescribers towards this challenging topic. There could be complex confounders such as time, resource, training, education, knowledge acquisition and cultural attitudes which might have influenced these response rate.’
Furthermore, since in general in the Netherlands there are large problems in recruitment of participants for studies in the field of intellectual disabilities and considering the costs and time-consuming nature of interview-studies, we doubt whether such a study currently could be feasible. Further, there is no obvious guarantee that a change in methodology would result in improved engagement especially if the root concerns are stigma and ignorance. However, the reviewer is correct in seeking other methods and we have put that as a possible approach for future consideration.
Finally, we agree with the reviewer that because of the low response rate the results of this study may not be representative for the study population (i.e., prescribers of psychotropic drugs to people with intellectual disabilities [ID] in the Netherlands). We have included this statement in the limitation section.
The available results clearly indicate that the adherence of prescribers to antipsychotic drug prescribing guidelines in ID mental healthcare needs improvement and the study also provides insight in barriers and facilitators for this improvement. On these grounds we believe we have a reasonable case for this paper to be considered for publication. Not doing so would perpetuate the ignorance and challenges of this topic, the very issues the reviewer is concerned for. We thank the reviewer for the opportunity to think deeper on this issue.
Other deficits in the manuscript as mentioned by reviewer 1:
1) The introduction section should be streamlined so that the authors express their main points in a more concise manner. For the topic at hand, a single page should be sufficient.
Answer:
We have now shortened the Introduction section to approximately one page by removing redundant text fragments.
2) The manuscript should be edited by a native speaker of the English language to correct numerous awkward sentences, repetitive phrases and incorrect punctation. Also, for the sake of consistency, the acronym of the Dutch association of psychiatrists should be provided.
Answer:
Thank you for highlighting this. On reflection we accept this criticism. Two of our co-authors who are from the UK have now proof read the paper and made considerable edits to the language style, syntax and grammar. Specific examples include:
- Removal of repetitive phrases, e.g., the term ‘challenging behaviour’ where this did not lead to possible misunderstandings of the sentences.
- We changed the alphabetic order of the five themes of the questionnaire in the Methods section/Materials in a numbered list:
‘The 44 questionnaire items were grouped under five themes similar to those in the UK survey. These themes included:
1) data on working experience and antipsychotic prescription practice in patients with ID.
2) experience in deprescribing antipsychotics.
3) how their antipsychotic prescribing practice including deprescribing, changed over the past five years.
4) how in the past five years new Dutch Act and guidelines influenced antipsychotic deprescribing.
5) potential facilitators and barriers in achieving the goal of deprescribing.
- We have now added the acronym of the Dutch Psychiatrists Association (NVvP) in the section Methods/Procedures:
‘… psychiatrists affiliated to the Dutch Psychiatrists Association (NVvP)…’
3) The authors should clarify the side-effects that are causing concern and point out that a cost-benefit analyses is always required to weigh the negative effects of a medication against its positive effects prior to prescribing. Many medications other than antipsychotics are prescribed off-label due to an acceptable cost-benefit analysis.
Answer:
The reviewer is right that the expected pros and cons of psychotropic drug prescription should always be weighed before a treatment starts.
We changed the text fragment in the Introduction section by adding information about the side-effects and about the conditions which should be met before prescribing psychotropic drugs for challenging behaviours in people with ID:
‘Moreover, in this population the risk for side-effects like diabetes, cardio-vascular disorders, sedation and movement disorders is considerable, especially in case of long-term use of antipsychotics [28-32]. It is recommended by all good practice guidelines that the treatment of challenging behaviours should be, in the first place, non-pharmacological, i.e., psychosocial and behavioural. Psychotropic drugs may only be prescribed after careful weighing of the potential positive and negative effects of this medication and preferably just for short-term (max. 6-12 weeks). For example, prescription may be beneficial to overcome crisis situations when the person or others are at risk of serious harm, or in conjunction with non-pharmacological interventions for treatment of severe symptoms of comorbid autism [16,33,34].
4) The entire section entitled Setting does not belong in the Methods section. Typically, a Methods section includes Participants, Materials, and Procedures. The details about the training of ID specialists does not seem particularly relevant to the present study, but when it is, it should be placed in the Introduction or Discussion.
Answer:
We thank the reviewer for this comment. We have moved the description of the health care services for people with ID to the introduction.
We have also ordered the subheadings of the Method section as follows:
Design
Participants and settings
Materials
Procedures.
Second, we accept the reviewer’s feedback on prescriber training is not particularly relevant to the present study. So, we removed redundant information about the ID physicians’ specialist training according to the reviewer’s suggestions. We also removed the sentence that there is no ID specialized training for psychiatrists and specialist nurses in the Netherlands.
5) The authors should explain why they consider it ‘surprising’ that patients’ own coping styles, the existence of supportive environments, and a good relationship with their doctor, was a key factor for successful discontinuation. It seems to me that this finding should have been expected.
Answer:
The author is right regarding the latter two points. However, there are not many studies who address the involvement of the antipsychotic drug users with a mild ID in the discontinuation process; this is mentioned in the discussion section’(see below). Therefore, the finding of the influence of users’ own coping style on achievement of complete discontinuation is perhaps not surprising, but important. We changed the sentence by adding ‘next to’ and by replacing the term ‘surprising’ into ‘important’:
‘Also, often individuals with ID themselves lack information and/or their voices are not heard in decisions in prescription of antipsychotics [43,56]. Previous studies of interviews of adults with ID have just touched on the issue of psychotropic medication use [57,58]. To address this knowledge gap, in another study we also gathered information on the experience of adults with a mild ID who have gone through the experience of psychotropic withdrawal [59]. An important finding was their statement that their own coping style, next to a supportive environment and a good relationship with their doctor, was a key factor for successful discontinuation.
6) The Discussion section should be significantly abbreviated so that the main points are more concisely discussed. As an example, the detailed description of the work by Deb et al. is unnecessary to make major points about the findings within the present investigation.
Answer: we shortened the discussion section and removed the detailed descriptions of the work of Deb et al.
Finally, the reviewer recommends the present study to consider as a pilot study as a rationale to run a more comprehensive assessment of the study topic, prior to seeking publication.
Answer:
We agree with the reviewer that the present study may be considered as a pilot study and should be followed by further research in the topic. Yet, we think that publication of this pilot study could be very helpful in the recruitment of participants in such a future study by increasing the interest and attention of prescribers/clinicians in the topic. This will likely stimulate their willingness to participate in such a study to contribute to knowledge about good clinical practice in psychotropic drug prescribing to people with ID.
Reviewer 2 Report
Thank you for inviting me to review the paper entitled “Withdrawing antipsychotics for challenging behaviours in adults with intellectual disabilities; experiences and views of prescribers”. This is an interesting survey about real-life experiences of discontinuing antipsychotics in people with ID and challenging behaviors. Given the lack of guidelines on the psychopharmacology of ID, the topic is relevant for both clinicians and researchers.
There is one major limitation, which unfortunately cannot be overcome at this point, i.e. the number of respondents to the survey is very low. Other comments:
Introduction
- The authors may cite the paper by Fusar-Poli et al. (doi: 10.1016/j.psychres.2019.04.013) about the prescription of psychotropic medication in adults with ASD, in which ID is regarded as one of the main characteristics assciated with antipsychotics prescription
Results
- First question (Table 1): It would be interesting to know how many patients were in charge of each respondent.
- The survey was made up of 44 questions, but only a few of them were actually reported in the paper. Please, explain or expand the results accordingly.
- Any difference between first- and second-generation antipsychotics?
Discussion
- In the discussion, the authors focus on a comparison with the survey conducted by Deb et al. Are there similar experiences in other countries?
- Do you think there might be potential fears about legal issues when deprescribing antipsychotics?
Author Response
Dear Editor and Reviewers,
Please find our revised manuscript entitled ‘Withdrawing antipsychotics for challenging behaviours in adults with intellectual disabilities; experiences and views of prescribers’; Manuscript ID: ijerph-1992619.
We thank the reviewers for their comments and questions which we have tried to address and answer as best as possible. We think that the clearness and readability of the manuscript has been improved by following the reviewers’ suggestions. The quality of the manuscript has improved by the shortening of the introduction and discussion section, a better explanation and lay-out of the tables and removing and adding of some text fragments. We have also paid attention to the writing style, which we hope now comes across as being better.
Please find below our specific replies to the reviewers’ comments.
Reviewer 2
We thank the reviewer for the words of appreciation regarding the relevance of our study for clinicians and researchers. Indeed, the low response rate is a limitation which we could not overcome in this study, and this is also mentioned in the discussion section.
Introduction
The authors may cite the paper by Fusar-Poli et al. about the prescription of psychotropic medication in adults with ASD, in which ID is regarded as one of the main characteristics associated with antipsychotics prescription.
Answer:
We referred to this publication in the Introduction section by adding the sentence:
‘Significantly higher levels of antipsychotic drug prescribing occur to manage challenging behaviour in people with ID and/or autism [23].’
We added
‘[23] Fusar-Poli L, Brondino N, Rochetti M, Petrosino B, Arillotta D, Damiani S, Provenzani U, Petrosino C, Aguglia E, Politi P. Prevalence and predictors of psychotropic medication use in adolescents and adults with autism spectrum disorder in Italy: A cross-sectional study. Psychiatry Research 2019, 276. Doi: 10.1016/j.psychres.2019.04.013’
to the reference list.
Results
- It would be interesting to know how many patients were in charge of each respondent (table 1).
Answer: Unfortunately, we have no data about the number of patients with ID the respondents were responsible for.
- The survey was made up of 44 questions, but only a few of them were actually reported in the paper. Please, explain or expand the results accordingly.
Answer:
We agree with the reviewer that the report of the results of the survey/questionnaire may seem incomplete. However, a lot of questions must be seen in conjunction with each other and therefore the answers could better be ordered according the five themes of the questionnaire in two clusters, just highlighting the main points. We added this sentence to the Results section. Moreover, for the completeness of information about the Materials, we added the items of the questionnaire and corresponding answer categories in Appendix 1:
Methods section:
‘The questionnaire items were adapted for the Dutch ID healthcare system from a survey which was developed in a recent study of UK psychiatrists’ views on the rationalization of psychotropic use among people with ID [44]. Answer options of items were multiple choice and open field. Appendix 1 shows the items of the questionnaire and corresponding answer options.’
Results section:
‘We present the results by the forming of two clusters encompassing the five themes of the questionnaire as described under Methods/Materials (theme 1 and 2, and theme 3, 4 and 5 respectively). The data and analyses of all the 42 questions and corresponding answers (see appendix 1) are available on request.’
- Any differences between first-and second-generation antipsychotics?
Answer:
Unfortunately, we have no data to answer that question.
Discussion
- In the discussion, the authors focus on a comparison with the survey conducted by Deb et al. Are there similar experiences in other countries?
Answer: Unfortunately, there are no studies from other countries about experiences of prescribers in ID healthcare. There are several studies on this topic carried out in dementia care among nurses and users of psychotropics, and one among nurses and support professionals in ID healthcare (Kleijwegt et al. 2019). We added a sentence in the Discussion section:
‘As far as we know there are no studies from other countries reporting prescribers’ experiences with antipsychotics discontinuation in patients with ID.’
- Do you think there might be potential fears about legal issues when deprescribing antipsychotics?
The reviewer addresses an important, but difficult issue. Indeed, with this new Dutch Act on Involuntary care, in which psychotropic drug prescription for challenging behaviours is judged as a restrictive measure, legal issues play a role. However, in some conditions clinicians are allowed to deviate from this Act and not to follow the guideline recommendations.
We added these sentences to the discussion section:
‘Legal issues may play a role. We found that 70% of the participants in our study indicated that the new Dutch Act on Involuntary care, aimed to reduce restrictive practice, and the new Dutch guideline on problem behaviour in adults with ID had encouraged their service providers to facilitate psychotropic drugs deprescription. Yet, clinicians are allowed to deviate from this Act and from the Dutch guideline on problem behaviour provided that they can substantiate this decision.’

Reviewer 3 Report
The paper by Dekuijper et al. describes a cross-sectional study aimed at evaluating attitudes and success with reducing or terminating the use of antipsychotic drugs in patients with intellectual disabilities (ID) for challenging behaviors. This is an important topic in view of both the side effects of long-term use of antipsychotic drugs and recent changes in national guidelines for reducing off-label use of these medications. Questionnaires were sent to physicians who treat patients with ID and psychiatrists and nurses who specialize in the care of this population. Overall, there was a 10% response rate, which is adequate. The responder population was diverse enough to provide representative insights into the prescribing behavior of healthcare professionals involved in the care of ID individuals. The methods and analysis were appropriate and very useful data were obtained. The authors also identified barriers and facilitators for reducing prescription of antipsychotic drugs. The findings are meritorious, but attention to a few issues might further strengthen the paper.
1. It was unclear how the numbers in Table 1 add up. In the text, it mentions that 9 respondents – 5 reporting 50-75% and 4 reporting 75-100% – had more than 50% of their ID patients on antipsychotics. Yet, in line 4 of the Table under Attempts to Withdraw APDs, the number was 18 (53%). It seems to me that this number should be 9 as a maximum, given the above, but I may have missed something. In line 7 of this Table, the number is 22 and this also seems higher than it should be. How were these numbers generated? Some clarification would be helpful.
2. In the Results section, lines 221-223, the authors appear to be comparing groups and provide numbers such as 9% versus 8% and 24% versus 77%. I think the second number refers to the psychiatrists/specialty nurses, whereas the first number in the pair refers to ID physicians. This wasn’t explicitly stated. Sorry for my confusion here.
3. In the Complications part of Table 2, the relationship between the numbers on the right and the items on the left is unclear to me. Better alignment of the text and numbers or further explanation may clarify this point.
Author Response
Dear Editor and reviewers
Please find our revised manuscript entitled ‘Withdrawing antipsychotics for challenging behaviours in adults with intellectual disabilities; experiences and views of prescribers’; Manuscript ID: ijerph-1992619.
We thank the reviewers for their comments and questions which we have tried to address and answer as best as possible. We think that the clearness and readability of the manuscript has been improved by following the reviewers’ suggestions. The quality of the manuscript has improved by the shortening of the introduction and discussion section, a better explanation and lay-out of the tables and removing and adding of some text fragments. We have also paid attention to the writing style, which we hope now comes across as being better.
Please find below our specific replies to the reviewers’ comments.
Reviewer 3
We appreciate the reviewer’s judgement regarding the importance of the topic of our study and the sufficient methodology of our study. However, there was some unclearness in the presenting of the results which we tried to address as described below.
- It was unclear how the numbers in Table 1 add up.
Answer: the reviewer is right regarding the complexity of the table and the description of the results in the text. However, the results section starts with the description of the prescription practice of participants, i.e., the number of participants who indicated that XX % of their patients were prescribed antipsychotics. Table 1 summarizes the deprescription policies, i.e., the number of participants who indicated that YY% of their patients were deprescribed antipsychotics. We added two sentences to explain table 1 and hope now it will be clearer.
‘Participants attempted to deprescribe antipsychotics in a variable part of their patients. In table 1 the number and percentage of participants is shown who attempted to deprescribe antipsychotics in a certain proportion of their patients.’
- In the comparison regarding the mean duration of the discontinuation trajectory in ID physicians versus ID mental healthcare professionals (psychiatrist and specialist nurses) it was not explicitly stated which percentage belonged to which group.
Answer:
We apologize for this mistake and have added this information in the text:
‘(0-6 months 9% in ID physicians versus 8% in ID mental healthcare professionals, 6-12 months 24% versus 77%, 12-18 months 47% versus 0%, and >18 months in 19% versus 15%, respectively, p=0.01)’
- In table 2 the relationship between the numbers on the right and the items on the left is unclear. Also, the alignment could be better.
Answer:
We added the extra information at the top of the right column and in some cells of the left column.
Also, we have restored the alignment in a part of the rows.
Thank you all
Round 2
Reviewer 1 Report
I appreciate the authors' attention to the issues I raised upon first review of this manuscript. The shortening of the both the Introduction section and Discussion section is quite an improvement. I still have concerns over the low response rate; however, the authors have at least called more attention to this issue in the Limitations section, and this is important and appreciated. I remain somewhat concerned over the fact that there was no face-to-face follow-up component in the study, and this is the reason for my lower rating on scientific soundness. Nevertheless, the improvements in the manuscript and the authors' qualifications of their results are such that I will recommend publication once a few minor edits are made (such as removal of "[GdK1][SR(PNFT2]" text on page 4 and removal of similar errors elsewhere). Thank you and best wishes.
Author Response
Dear reviewer,
Please find our revised manuscript entitled ‘Withdrawing antipsychotics for challenging behaviours in adults with intellectual disabilities; experiences and views of prescribers’; Manuscript ID: ijerph-1992619 revised version. We thank the reviewer for his/her approval of the changes we made in the original manuscript according to the comments and suggestions. However, some minor changes in the revision were still needed which we hope are addressed adequately in this second revision.
The reviewer suggested some minor edits regarding the English language and to check the spelling.
You will find the spelling and grammar changes shaded in yellow in the main document. We also removed the comments in the margin.
We hope that the manuscript is suitable now for publication and look forward to the decision of the editors.
Yours sincerely, on behalf of the co-authors,
Gerda de Kuijper
Reviewer 2 Report
The authors have responded to the comments raised satisfactorily.
Author Response

(The authors gave the same response as above.)
